# Using Walk-Along Interviews to Identify Environmental Factors Influencing Older Adults’ Out-of-Home Behaviors in a High-Rise, High-Density Neighborhood

**DOI:** 10.3390/ijerph16214251

**Published:** 2019-11-01

**Authors:** Yuxin Cao, Chye Kiang Heng, John Chye Fung

**Affiliations:** 1Center for Ageing Research in the Environment, School of Design and Environment, National University of Singapore, Singapore 117566, Singapore; 2Department of Architecture, National University of Singapore, Singapore 117566, Singapore

**Keywords:** age-friendly neighborhood, out-of-home behaviors, walk-along interview, high-rise, high-density

## Abstract

Older adults’ out-of-home behaviors (OOHBs) are critical for maintaining health and quality of life. Taking Singapore’s Yuhua East as a case, this study applied a qualitative approach to explore what neighborhood environmental factors influence older adults’ OOHBs. Twelve older adults were recruited for walk-along interviews through the use of purposeful convenience sampling. A content analysis was conducted using NVivo 11 via an inductive approach. Research results revealed 12 categories of environmental factors that affected older adults’ OOHBs: access to facilities (shops and services, public transit, and connectivity), pedestrian infrastructure (sidewalk quality, sheltered walkways, universal design, crossings, benches, and public toilets), aesthetics (natural elements, buildings, noise, and cleanliness), traffic safety (behavior of other road users and road width), safety from crime, wayfinding, familiarity (long-term residency and routine activities), weather, social contact, high-rise, high-density (lifts, population density, flat size, and privacy), affordability (shops and services, as well as transportation), and maintenance and upgrading. This analysis concluded that access to facilities and pedestrian infrastructure are important for older adults’ OOHBs. Considering Singapore’s weather, sheltered walkways, the proximity of facilities and connectivity should be given serious emphasis. In addition to physical factors, social contacts and the affordability of shops and services are also important.

## 1. Introduction

Out-of-home behaviors (OOHBs) have been demonstrated to be important for older adults’ health and quality of life. OOHBs refer to engagement in activities out-of-home as well as the full range of behaviors of moving from one location to another [1,2]. Though there have been studies exploring the relationship between OOHBs and several socio-demographics, cognitive and physical functions, little emphasis has been put on environmental factors. According to environmental docility theory [3], the more incompetent the person, the more dependent they are on environmental factors. The neighborhood environment becomes more and more important for OOHBs in older adults.

A neighborhood environment is both physical and social [4]. There have been a number of quantitative articles exploring environmental effects on older adults’ OOHBs, mostly focusing on physical activities such as walking, and the research results have been inconsistent. A review of quantitative studies exploring the relationship between the physical environment and physical activities in older adults showed inconsistent results, and most of the physical characteristics (e.g., access to services and land use mix diversity) were found to be insignificant [5]. Another quantitative study review article concluded that there was sufficient amount of evidence showing physical environment characteristics (e.g., residential density and access to destinations/services) are positively related to older adults’ active travel, and there was a positive association between neighborhood walking and pedestrian-friendly features and availability of benches/sitting facilities [6]. These inconsistencies may be due to the fact that current quantitative studies usually look into a few characteristics of the physical environment and oversimplify the complex ways in which the physical environment impacts people’s physical activities [7]. Physical activities alone do not provide a holistic picture of older adults’ daily living experiences. These studies also do not take neighborhood social environment into consideration.

Meanwhile, there have been a small number of studies using qualitative approaches to explore the impacts of the environment on older adults’ OOHBs, especially using spatial qualitative methods. One review article looked into qualitative studies investigating physical environment and physical activities, and it identified 10 articles, three studies using photovoice, three using walk-along interviews [8,9,10], and three using observation after or before interview [11]. Though more spatial qualitative studies have emerged since 2014 [12,13,14,15], they have been conducted in western countries with predominantly sub-urban populations. Few studies have looked into older adults’ OOHBs (besides physical activities) in the high-rise, high-density context. Therefore, this study took a qualitative approach to explore how neighborhood physical and social environment encourages and/or inhibits older adults’ OOHBs in high-rise, high-density neighborhoods in Singapore. Singapore is a highly urbanized Asian city-state where more than 80% of the residents live in high-rise, high-density public housing estates developed by the Housing and Development Board (HDB). It is a rapidly ageing country the proportion of residents aged 65 years and over is 13.7%, and the proportion of those aged 55 and over is 28.1% [16]. This study used a walk-along research method due to the complexity of person–environment relationships and intends to get detailed and place-based information.

## 2. Materials and Methods 

### 2.1. Study Area

The Yuhua East neighborhood is located in the town of Jurong East, in the west region of Singapore (Figure 1). It is a typical example of HDB new town developments in late 1970s and 1980s, following “New Town Structural Model” with amenities planned at three hierarchies from a town center, to neighborhood centers, and then precinct centers (see [17] for details about HDB new towns). Defined by administrative boundaries, Yuhua East covers a 93-hectare area of land with a population of 26,330, where 17.8% of the residents are aged 65 years old and over and 34.8% are 55 years old and over [16]. Yuhua East is bounded by the Pan Island Expressway, Jurong Town Hall Road, and Jurong East Central. The Yuhua West and Toh Guan neighborhoods are located to its west and east, respectively. A 13-hectare park named Chinese Garden is located to its southwest, and the town center with shopping malls, Jurong Regional Library, and the bus interchange are located to its southeast. Housing type is a key symbol of social-economic status in Singapore. In Yuhua East, 88.5% of the residents live in public housing estates, with 37.3% living in three-room flats, 30.6% live in four-room flats, and 32.1% live in five-room and executive flats [16]. The residents here are 73.4% Chinese, followed by 13.6% Malay, 10.9% Indian, and 2% other. Though data are not available for Yuhua East, among those aged 55 years old and over, the proportion of Chinese is higher at 80.9% [16]. This article focuses on the public housing area within Yuhua East. 

### 2.2. The Walk-Along Method 

The major advantage of conducting walk-along interviews is that it allows researchers to observe participants’ spatial behaviors, access and interpret their perceptions and experiences about the surrounding environment when the researchers walk together with the participants, ask questions, and listen along the way [18,19]. In spite of their advantages, walk-along interviews may be influenced by conditions that are outside the control of the researcher, such as the weather [20]. The presence of the researcher may also disturb and intrude the participants’ ordinary behaviors and influence the participants’ walking experiences [18]. For studies using walk-along techniques, some have taken a natural approach in which researchers go together with the participants and chat in their familiar areas with no expected routes and durations [18], while some take a structured approach with pre-determined durations, destinations, and walk-along routes [8]. The degree of the researcher’s involvement in the walk-along interview varies depending on the purpose of the research [12]. 

To gather more information and empower the participants as much as possible, this study took a more unstructured approach. The participants were recruited at amenities and open spaces within the boundary of Yuhua East. After explaining the research purposes and procedures to the participants, the researcher walked together with the participant and asked questions along the way from a neighborhood destination to the common corridor of the participant’s home, or the other way around. The walk would stop or begin, normally at void decks (vacant spaces on the ground floor of HDB flats) if the participant felt uncomfortable. In this case, information about the missing parts of the route would be collected without walking-along. This study allowed the participants to talk as freely as they liked even though the conversation went beyond the study purpose. This also allowed the participants to conduct various activities along the route or stop for a chat. As a result, the duration of the interview ranged from 9 min to more than 2 h.

### 2.3. Data Collection and Analysis

Walk-along interviews were carried out from August to September 2018 by one researcher using purposeful convenience sampling. The researcher’s mother language is Mandarin Chinese, with Test of English as a Foreign Language^®^ (TOEFL^®^) test score of 106. Though older adults are generally defined as those above 65 years and over, this study included those aged 55–64 because that Singapore is ageing rapidly and their opinions are valuable for future developments. The percentage of participants aged 55–64 was expected to be less than 20%. Selected samples were also expected to cover similar numbers of male and female, including Chinese, Malay, Indian, and other. To be recruited, participants needed to be able to speak either English or Mandarin Chinese, living in Yuhua East, or living in the nearby neighborhood but walking to the facilities in Yuhua East on a daily or weekly basis. Interviews were recorded and transcribed for analysis if the participant agreed. Otherwise, the researcher jotted down key phrases and facts on the spot and expanded them into full sets of field notes as soon as possible after the interview. During the walk, the researcher also took photographs of the things mentioned. The routes were hand drawn on printed maps, compiled, and digitalized later. The starting/ending points on the digital map were adjusted to the surroundings to protect the participants’ information. The study protocol was approved by the IRB of the university (S-18-155E). 

Data from the transcribed interviews, field notes and photographs were added to the NVivo software. The walk-along interviews were conducted by the researcher in English (*n* = 4) and Mandarin (*n* = 8). Field notes and transcribed interviews were in their original language, and the translation was done by the same researcher who conducted the interviews. The translation took place during the analysis to ensure the authenticity of the findings [21]. The data were analyzed using content analysis via an inductive approach [22]. Categories and sub-categories were derived based on previous studies [6,8] for ease of comparisons. 

## 3. Results

### 3.1. Participant Information and Walk-Along Routes

The final study sample consisted of 12 participants (six male and six female), between 55 and 80 years old, including 10 Chinese, one Indian and one other (Table 1). The researcher failed to recruit Malay participants, although attempts were made. A higher proportion of participants live in five-Room HDB flats, probably because older adults tend to have larger household sizes. Figure 2 illustrates the routes of the walk-along interviews. 

### 3.2. Content Analysis

A qualitative data analysis revealed 12 categories of environmental factors that affected the older adults’ OOHBs (Figure 3): access to facilities (shops and services, public transit, and connectivity), pedestrian infrastructure (sidewalk quality, sheltered walkways, universal design, crossings, benches, and public toilets), aesthetics (natural elements, buildings, noise, and cleanliness), traffic safety (behavior of other road users and road width), safety from crime, wayfinding, familiarity (long-term residency and routine activities), weather, social contacts, high-rise, high-density (lifts, population density, flat size and privacy), affordability (public space and services as well as transportation), and maintenance and upgrading. The numbers and percentages of participants that discussed an environmental category can be seen in Table A1 in the Appendix A. By taking an unstructured approach, some important categories were uncovered that went beyond environmental factors, including social norms (social attitudes towards ageing, gender difference, and multi-ethnic country), government policy, and health.

### 3.3. Access to Facilities

#### 3.3.1. Shop and Services

All the participants (*n* = 12) mentioned that shops and services along the way encourage them to go out, including a hawker center (an open-air complex housing many stalls selling affordable food), a market, coffee shops, supermarkets, exercise corners, parks, kindergartens, playgrounds, schools, a community club and a social service center. They also mentioned shops and services not seen along the way that entice them to go out, such as hawker centers in other neighborhoods and Jurong West (another town next to Jurong East), a Chinese Garden, polyclinics, National Trades Union Congress (NTUC) LearningHub (for skills development), Jurong Regional Library and shopping malls located in the town center. 

The proximity of shops and services influences older adults’ transport modes. For example, a male participant usually walks to Jurong Regional Library but takes public transport to the polyclinic. To reach the facilities that are necessary for daily living, older adults are willing to tolerate longer distances. A female participant with a walking aid said that she takes about an hour to go to the market and come back. Having a sick husband at home and hoping to cook dinner for the children, going to the market is the only out-of-home behavior she performs. She does not mind that the distance is a little bit far since she considers it as an exercise. The tolerance for proximity reduces for recreational destinations such as parks. A participant in his 80s said that he visits the market twice a day. However, for the park located next to the market, the participant said: “I am getting old now, I do not go there anymore. I went there before.” Meanwhile, long commute distances may inhibit the employability of older adults. A female participant who works part-time mentioned that it is not worth it for her friend to take two hours to commute but only works for three hours. 

Besides proximity, the quality of services also attracts the older adults. For example, although there is a hawker center and market in Yuhua East, the participants mentioned another one in Jurong West because it has a variety of food and shops. 

#### 3.3.2. Public Transit

Many participants (*n* = 9) mentioned access to public transit. For those who are still working or occupied by various interests, living near to the Mass Rapid Transport (MRT) stations and bus stops are important for their OOHBs. When asked about her feelings about her neighborhood, a participant stated: “This area is very good actually. You can go everywhere, it is easy. If you want to go to Jurong East MRT, very convenient, you can walk or take buses. I have stayed here for 18 years.” Another participant specifically pointed at the MRT station and said: “We are so close to the MRT. You see that there is an MRT station over there… Chinese Garden MRT. It is blocked by these trees and it is only an eight minute walk. Do you see the roof? That is Chinese Garden Station (Figure 4).”

#### 3.3.3. Connectivity 

A few participants (*n* = 3) mentioned a variety of pedestrian routes and multiple connections that permit flexible paths. A participant in her 70s mentioned that the increased connectivity gives her the opportunity to choose a path suitable for her physical capacity and the weather conditions:
“I want to take a short cut from here. Actually, you do not have to cut through here. You can walk on the main road and take the stairs down, then you go straight. But I want to avoid the stairs, so I go by this way. If it is raining, I do not walk here. I walk over there. It is a covered walkway all the way back. If you do not carry an umbrella, it is still okay.” 


Increased connectivity also promotes the convenience of walking to amenities. Participants expressed that the newly added sheltered crossing connecting the residential area and the neighborhood center improves their walking experiences. One participant said: “Now they do this (covered walkway); it is good. It is covered from the rain. If not, you must walk over there. You see? (Figure 5).” 

### 3.4. Pedestrian Infrastructure

#### 3.4.1. Sidewalk Quality 

Many participants (*n* = 10) were generally satisfied with the sidewalks because they are smooth, very easy to walk on, clean, and maintained regularly. Nobody complained about the continuity of sidewalks. One female participant said that the upgrading carried out by HDB did a good job in changing the sidewalk material from cement to tiles. She complained that one particular material turned dirty very easily. There were also participants who mentioned that the sidewalks tended to be slippery after rain and maintenance work (the flushing of the floor). The floor should be replaced by coarse materials. 

#### 3.4.2. Sheltered Walkways 

Some participants (*n* = 8) mentioned sheltered walkways and considered them to be one of the most important features that make walking pleasant. When asked about the general feelings about a route segment, one participant said: “This route? Very good. People will not get wet if it rains.” Another participant was very satisfied with the extent of the coverage of the sheltered walkways and said: “We have covered walkways all the way to my place. Whether it rains or shines, I will be protected.” 

#### 3.4.3. Universal Design 

The majority of the participants (*n* = 8) discussed universal design during the interviews. Though some mentioned that stairs with only one step are dangerous and inconvenient, ambulant participants usually do not mind a few steps. Even for the participants with reduced mobility and using walking aids, rails can enhance their out-of-home mobilities. For longer stairs, the participants prefer ramps or elevators if possible. Though the neighborhood is wheelchair accessible, the provision of accessible routes does not take weather conditions and ease of travel into full consideration. A male participant, who has volunteering experiences of pushing wheelchairs, mentioned that the width of some ramps is too narrow, so they are very dangerous, especially at the turning point. The wheel may get stuck at the edge, and then both the wheelchair and the older adult on it may even roll over. A conversation during an interview with a participant and her friend also revealed the inconvenience of some accessible routes. 

Friend: “If this is without stairs, it will be good. Do the flat one. It is more convenient for people pushing the wheelchairs.”

The participant pointed to the slope a few meters away and said: “There is a slope.”

Friend: “Although there is, people will get wet in the rain. Here (the sheltered walkway), you do half stairs, half ramp, then people can push the wheelchairs. Although they say there is a slope, that is not covered (Figure 6).”

For overhead bridges, the participants’ opinions were influenced by physical capacities. Older adults with good physical capacities stated that they do not mind the absence of ramps. However, for participants with foot pain or ankle problems, ramps are necessary. For example, a participant said: “If there is the overhead bridge, I walk on the long ramp. I do not want to climb the stairs. My foot starts to hurt (Figure 7).” As the ramp is quite long, one participant mentioned that it is better to have elevators. 

#### 3.4.4. Crossings

The majority of the participants (*n* = 11) mentioned crossings. They felt that it is safe to cross the street if there are traffic lights. The presence of zebra crossings at school zones is considered very safe. As for waiting time, some participants do not think it is too long because they are not busy or in a hurry, while one participant complained about the waiting time and that sometimes people have to wait under the strong sun. For the green man time, the participants generally feel that it is enough. However, a participant in his 60s said that 35 s is necessary for an eight-lane road. He is satisfied with the green man time in Yuhua East, but feels that one crossing in Clementi (another town) has an inadequate green man time:
“You see that traffic light (in Yuhua East) has extra seconds now. I visit the temple in Clementi every week. For that traffic light, they did not extend the time. You count how many lanes here, one, …, eight lanes. For pedestrians to cross, you need to have 35 seconds. There, I have seen a few times, older adults do not have enough time to cross. For us, it is okay. But for older adults, 30 seconds is not enough time to cross the eight-lane road.” 


#### 3.4.5. Benches

A few participants (*n* = 4) mentioned benches. A male participant who walks 10,000 steps every day stated that he did not perceive the need for providing seats along the way: “I seldom sit. I take a rest after I finish, not halfway.” A seating area not only serves as a resting spot but also as a place for social gatherings. A participant mentioned a seating area in the void deck and said that some blocks have only a few benches:

“They (HDB) do this. It is their small meeting garden. You can sit down and chit-chat. If you are tired, you can also rest. They (other residents) said several blocks do not have enough benches for them to sit. For example, when you wait for a taxi or other transports, there are not enough places to sit, so they have to stand (Figure 8).”

#### 3.4.6. Public Toilets 

Very few participants (*n* = 1) mentioned public toilets as a factor influencing OOHBs. They knew that public toilets can be found in shopping malls and coffee shops.

### 3.5. Aesthetics

#### 3.5.1. Natural Elements

Some participants (*n* = 6) expressed that they enjoy greeneries during their walks. For example, one participant stopped at a park (Figure 9) and said: “It is very good. You can see for yourself. You can take a picture, very nice environment.” One participant mentioned the healing effects of being in touch with natural elements:
“While walking, I am trying to understand all these plants. When you observe nature, if possible, you touch these plants, you feel healthier. For some reason, because you are under the sun, you are at one with the plants. In farming or gardening, there are some therapeutic elements. To me, the importance of gardening is actually beyond the physical activities.” 


#### 3.5.2. Buildings

Some participants (*n* = 2) mentioned buildings. One participant mentioned that the clean buildings and roads make her walking experiences pleasant, and another mentioned that it is a pity that a lot of old buildings had been torn down. 

#### 3.5.3. Noise

A few participants (*n* = 3) talked about noise. One participant mentioned the noise from the construction sites, and another complained that the hawker center is sometimes very noisy. The participants expressed that they enjoy the quietness of the common corridors, which are shared by fewer flat units.

#### 3.5.4. Cleanliness 

Some participants (*n* = 4) expressed that the clean environment makes their out-of-home activities pleasant. The cleanliness is the result of regular maintenance. One female participant said: “The road is usually clean, sometimes there are litters before the washers do their work. A little bit of waste there but usually very clean. Everything I can feel, the buildings (and) roads are clean, so I feel comfortable.” 

### 3.6. Traffic Safety

#### 3.6.1. Behavior of Other Road Users 

Some participants (*n* = 4) expressed their concerns regarding traffic safety on the sidewalks and roads. As older adults may need extra green man time to cross the road, the participants mentioned that some drivers do take notice and slow down while others pay no attention. Regarding sidewalks, the participants expressed safety concerns related to the behavior of other road users, cyclists and those on personal mobility devices. A participant said: “Be careful. Sometimes, there are people who ride on scooters or bicycles. Need to be careful (and) walk on one side (Figure 10).” One male participant goes out for his walking exercise as early as 6:30 am to avoid the traffic and he said: “At 6:30, there is little traffic. The most horrible ones are those on e-scooters. They come near to you without making any sound. If they knock you down, they (would) just say I have no money to compensate you.”

#### 3.6.2. Road Width 

A few participants (*n* = 2) talked about road width. Local roads between blocks are perceived as safe. For wider roads, the participants wait at the traffic light to be safe. 

### 3.7. Safety from Crime

A few people (*n* = 2) mentioned safety concerns related to crime. The neighborhood was perceived as very safe. For example, a participant who used to be a security guard shared: “It is very safe to walk in Singapore. I told everyone there is no need to be afraid at night. I myself walk back and forth every week in the night. Unless you are very unlucky (to) meet those drunk men, very rare. Coming back at midnight, it is still safe.” 

### 3.8. Wayfinding

Some participants (*n* = 5) shared with the researcher how they find their ways to get to a facility mentioned during the interview and how they get back. Road junctions and specially built environment features usually help, such as overhead bridges, bus stops, schools, hawker centers and shopping malls. The participants also mentioned that a place with a large population is usually the place they want to go. 

### 3.9. Familiarity

#### 3.9.1. Long-Term Residency

None of the participants expressed problems with wayfinding. This is probably due to long-term residency. For example, one participant mentioned that he had lived in his block for more than 40 years and that the hawker center was built at the same time as his block. 

#### 3.9.2. Routine Activities

Routine activities conducted by the participants (*n* = 11) were also stated to improve familiarity and encourage OOHBs. For majority of the participants, routine activities generally involved eating, shopping, and exercising, while some participants also work, attend courses, and attend group activities. A participant who walks back and forth to IMM at the town center perceived no obstacles along the way: “I get used to walking. I walk every day. Climbing up the stairs and climbing down are the same. It does not matter.” For the participant on a wheelchair accompanied by a foreign domestic worker, her out-of-home routine activities are carried out as follows: being pushed to the exercise corner across the street in the morning and doing exercises by herself, then being pushed to the market and walking into the store, buying goods and food by herself, being pushed back to the void deck and meeting friends, and then finally being pushed back home. Her familiarity with the neighborhood helps her find accessible routes, locate shops necessary for daily living, gain a certain level of independence, and maintain friendships. 

### 3.10. Weather

Singapore has a typical tropical climate with high and uniform temperatures and with high humidity all year round. Many participants (*n* = 9) mentioned that hot weather or rain influences OOHBs, especially for recreational activities. The participants expressed that they avoid going out in the early afternoon and prefer to do just a little exercise if the weather is nice. For the participants who have jobs, join fixed-time activities, or need to buy necessities, they go out regardless of the weather conditions. In this case, sheltered walkways play an important role in encouraging older adults to go out. Sheltered walkways with slopes are necessary for wheelchair users’ OOHBs because the caregivers may not have free hands to hold the umbrellas for them (Figure 11). Additionally, older adults need to be careful when they go out in the hot weather. A participant in his 80s shared his dangerous experience during a follow-up informal interview:
“Yesterday, my blood pressure went up to 180+. At 4:00 p.m., I walked back from the MRT (Chinese Garden). The weather was so hot and my skin sticky, very strong sun. That night, I took a shower and did a little bit of exercise. Then I felt my body was very hot, my eyes were foggy. I took my blood pressure and it was more than 180. It has never been this high before.”


### 3.11. Social Contacts

Social contacts are an important factor encouraging OOHBs in older adults. Four participants met their friends or neighbors during the interviews and said “Hi” or waved their hands to their friends. Some participants (*n* = 7) talked about their friends along the way, their friends drinking coffee in the hawker center, chit-chatting in the void deck, doing group exercises, and even their friends taking courses together. Maintaining social contacts is considered necessary for mental health, preventing depression and dementia. A participant shared a story about her son’s English teacher:
“After retirement, she did not contact anyone, very self-centered. Within half a year, she had depression, feeling that there was nothing to do. She might have thought that she was useless. I think she probably thought like this. If she had come out and learnt something, she would not end up like this. The important thing is to contact people, then you will not get dementia so fast.”


The participants also talked about social contacts made during volunteering or working. Social contacts are information sources and can potentially increase older adults’ employability. For example, one participant who quit her full-time job as a therapist due to neck pain shared that she got her current part-time job because her former colleague resigned and gave her the position. She said: “If no one resigns, there is no way to join.”

### 3.12. High-Rise, High-Density

#### 3.12.1. Lifts 

The participants generally stated that they generally do not perceive lifts as barriers for OOHBs because the lifts now stop on every floor and are maintained regularly. One participant said: “Our block has three lifts. The lifts are regularly maintained, Lift 1 being maintained, (Lifts) 2 and 3 (are) okay, so there is no problem.” Residents living in blocks undergoing upgrading may experience some inconveniences because only one lift may be operating to serve all the residents in that block, and the waiting time may be quite long. 

#### 3.12.2. Population Density

The participants stated that they do not perceive that there are too many people in the neighborhood, except in the hawker center, the market, and the public transit. One participant said: “On Saturday and Sunday, the hawker center is full. There is no place to sit. We need to wait until 9:00 a.m. or 10:00 a.m. to find a seat.” For older adults with good mobility, the high population in hawker center does not influence their visiting. However, it is a little inconvenient for older adults on wheelchairs to navigate. The foreign domestic worker of a participant said: “From Wednesday to Sunday, there (hawker center) are too many people. We are afraid to go in. Because she is on the wheelchair, it is not convenient. We go to the coffee shops nearby.” Another participant using a walking aid stated that she prefers to walk instead of taking the bus even though the walking distance is a little far. She explained: “Taking bus is not convenient. With this (a grocery trolley), you need to pull it onto the bus. Sometimes when there are too many people, I move very slowly.”

#### 3.12.3. Flat Size

Some participants (*n* = 4) expressed satisfaction with their spacious flats. Only one participant mentioned that he wanted to go out to take a break from the walls at home. 

#### 3.12.4. Privacy

The participants who mentioned common corridors (*n* = 3) said that they prefer quiet common corridors shared by fewer units. 

### 3.13. Affordability

#### 3.13.1. Shops and Services

The affordability of shops and services is considered a very important factor for older adults’ OOHBs. Public spaces that do not impose any charge for entry provide the older adults with a place to spend their time. For example, a participant said: “The library is where anyone can enter, you can read the newspapers or do other things.” Hawker centers and coffee shops with affordable food and drinks are ideal places for meeting up with friends or reading newspapers. People can stay for a whole afternoon or longer if the business is not too busy. A participant talked about his time spent in the hawker center and the need to save money:
“Hawker center is the best place. We can sit there whether there is business or not. If there is, we need to be sensible to give up the seats (to customers) after we have finished eating our food. They (stall owners) need to make money. If they (customers) go back to the office, you can sit for two-to-three hours, it does not matter then. It is free. In the hawker centers, you can find a meal for $2, three dishes, all vegetables. Meat is a little bit expensive (Figure 12).” 


The participants also mentioned the affordability of social services and courses. For example, a female participant who attends courses like qigong, yoga and Malay language said: “The price is affordable. We pay only $5 or $10 for 10 lessons. Qigong is free. Everyday exercise is free.” A 65-year-old participant shared her experiences in taking courses:
“… If our salary is below $2000 (per month), we can take courses and only pay 5% of the course fee. Like the computer course, which costs $500, we only pay about $20. Then, if you take two courses, they (the government) will give you $200. If you take four courses, they will give $400. I make money by taking courses.” 


#### 3.13.2. Transportation

Very few participants (*n* = 1) mentioned the affordability of public transport. One participant suggested that older adults, aged 65 years old and over, should enjoy free bus rides or pay only 50 cents per ride.

### 3.14. Maintenance and Upgrading

The majority of the participants (*n* = 9) talked about maintenance and upgrading, probably because Yuhua East is currently under the Home Improvement Program and Neighborhood Upgrading Program. Upgrading programs have positively contributed to the older adults’ OOHBs by improving sidewalk quality, having lifts that stop at every floor, adding pedestrian crossings and sheltered walkways, and updating exercise facilities. Meanwhile, the regular maintenance of blocks and common areas carried out by the town council guarantees the cleanliness of the neighborhood environment and the smooth operation of lifts. The participants generally gave positive feedback on what has been done: “This is very easy to walk, after they finish that (upgrading);” “This route looks very good. Last time, we do not have this, but now they have already upgraded everything.” Meanwhile, their requirements for the neighborhood environment are restricted by government programs and public expenditures. Their general comments were: “It is already like this, how to change it. I think it is impossible. There is no room to change. There are too many things. It takes a lot of money (and) it is not easy.”

### 3.15. Social Norms

#### 3.15.1. Social Attitudes towards Ageing

The participants’ feedback reflects social attitudes towards ageing. They discussed the vanishing of traditional value such as filial piety. Instead of being taken care of by their children, the participants expressed that they have to depend on themselves or the government. Older adults also think that the younger generations have very different opinions. For example, one male participant said: “For my age, it is okay. The young generation will probably think something better. My requirements are not high, it (the environment) is already good enough, I do not need anything more.” Some passive attitudes may be the reason that some older adults feel reluctant to go out, as shared by one participant who volunteers to befriend older adults:
“A lot of older adults do not want to learn. Many said why should they learn since they are dying, being so old. Now the government gives you money to learn something, such as cooking, in the community club nearby. They do not want to join. Some of them feel that it may be inconvenient, as they have foot pain, not being able to walk, but many older adults say that there is no point in learning as they have no one to cook for.” 


#### 3.15.2. Gender Difference

One male participant mentioned gender differences in OOHBs and the necessity to arrange activities that attract males:
“While walking, you can see some people do exercises, qigong, or whatever. Most of them are ladies. I do not see many ladies on wheelchairs. It is mostly men on wheelchairs. This kind of observation gives you some information that it has to do with the type of activities.” 


#### 3.15.3. Multi-Ethnic Country

As a multi-ethnic country, the provision of facilities in Singapore takes the different demands of the different groups into account. One participant from Myanmar expressed her appreciation for the different shopping choices in her neighborhood: “Here is an Indian shop. Usually Indians’ and Myanmars’ favorite foods are very similar. If I like to buy our food, I go and buy from the Indian shop.” Another participant expressed his pride about the tolerance of the hawker center for different ethnicities:
“Hawker center is great! Because no matter what meat you eat, what skin color you have, white or black, what religion you belong to, once you enter (hawker center), you can sit all together, there is no disagreement. I do not eat pork, but there is no need to separate us. You eat your food, sitting next to each other, he eats his.” 


### 3.16. Government Policy

While walking, the participants discussed government policies about housing, transportation and health. Though there were complaints about high healthcare cost, the participants appreciated the efforts of the government to promote OOHBs among older adults. For example, one female participant in her 60s said: “The government has provided many programs for the older adults to encourage them to go out of their homes.” Another male participant shared: “I cannot remember what it is called, national health… but it encourages people to walk 10,000 steps using certain measurement (pointing to the waist), like the steps tracker. The government is doing a lot of things, making the places friendly for older adults.” 

### 3.17. Health 

The participants’ health conditions influence their OOHBs. They generally perceived their health as good, but the majority of them have some health issues, including hearing problems, foot pain, knee pain, spinal pain, hypertension and high cholesterol. Those with foot and knee pains have difficulty climbing stairs. For example, one participant said: “For me, I do not mind going up the stairs, but going down is quite a problem.” Another participant expressed his safety concern because of his ear problems: “I cannot hear, (so I) do not even know who is coming from behind.” Confronted with these issues, the participants have adapted themselves to remain active by walking slowly and shorter distances and by replacing strenuous exercises with mild exercises. For example, one participant who had gone through cancer treatment said: “I do exercise every day, mostly walking exercise. It is very good for our age. Running or any other activities, no. When I was young, I played volleyball, badminton, ….” Meanwhile, the participants also mentioned that OOHBs are necessary for health.

## 4. Discussion

*Access to facilities:* Our findings suggest that good access to facilities encourages older adults’ OOHBs. This result is supported by previous studies [6,8,23,24,25]. This study has demonstrated that older adults have different tolerances for the proximities of different amenities. For those facilities that are necessary for daily living such as eating places and the market, the older adults would still visit even though the distance is a little bit longer. For recreational facilities such as parks, the proximity of amenities is very important [11,24]. Additionally, older adults are attracted not only by the facilities but also by the presence of other people, which is related to the sense of safety and opportunities for social contacts [8,26]. Studies have shown that older adults who visit parks have more concerns for safety because of the possibility of tripping and falling, as well as the fact also that older adults prefer to visit places with good visibility and presence of people who can help [27]. 

*Pedestrian infrastructure:* As proven in other studies [6,11,25], pedestrian infrastructure is an important factor influencing older adults’ OOHBs. In this study, none of the participants complained about sidewalk quality (e.g., uneven sidewalk or width) except for being slippery after the rain and maintenance work. Compared with a previous study [8], three sub-categories including universal design, sheltered walkways, and public toilets were added here. Sheltered walkways turned out to be the most important factor influencing older adults’ perceptions of sidewalk quality. This is probably due to Singapore’s tropical climate. Though the importance and principles of universal design have been well specified [28,29], in reality, the provision of accessible routes has minimal considerations for users’ conveniences and their abilities to use the space in the event of rain. Accessible routes usually involve longer travel distances. One-foot steps not only cause some inconveniences for wheelchair users but also increase the risks of falling for users. 

*Aesthetics:* The presence of natural elements and a quiet, clean and well-maintained neighborhood can encourage older adults to go out [6,8,24,30]. Compared to natural elements, buildings were rarely mentioned in this study. One participant lamented the loss of historic buildings, which were found to promote walking in previous studies [8].

*Traffic safety:* Road width was uncovered as a predicator for traffic volume and traffic speed. For another subcategory behavior of other road users, personal mobility devices have emerged as a safety concern for older adults [31,32], due to the device’s fast speed, small operation sound and flexibility. The concern becomes more prominent for older adults with hearing problems. Drivers’ behaviors also influence older adults’ feeling of safety at crossings. 

*Safety from crime:* Though it has been demonstrated in other studies as an important factor [8,30,33], few participants mentioned this environmental category in this study. This is probably due to Singapore being considered as one of the safest countries in the world [34]. 

*Wayfinding:* It was found that people use mental maps to help them navigate their ways in cities, and these maps are made up of five elements: path, edge, district, node and landmark [35]. In this study, older adults were found to use routes (walk along xx road), nodes (turn at the junction), and landmarks (e.g., overhead bridge, bus stop, and shopping mall) to find their ways. Due to the similarity of the HDB blocks, the block number is important for wayfinding. Participants also use the number of people as a marker since their destinations (e.g., polyclinic) usually have a lot of people, especially when taking buses. 

*Familiarity:* Familiarity can be explained by two sub-categories: long-term residency and routine activities. According to Rowles [36], older adults develop physical and psychological attachments to places after long-term residency. This “inside-ness” makes the older adults fully aware of the environment’s physical configurations, local services and social networks, and it promotes mobility and independence [36]. Continuity theory hypothesizes that older adults usually maintain their core attitudes, values, lifestyle activities, social relationships as they did in earlier days [37]. In this sense, routine activities can help promote OOHBs as older adults continue to grow old with reduced mobility. There is growing evidence showing that maintaining abilities to engage in routine activities and keeping an active lifestyle are beneficial for older adults’ independent living and health [38,39]. 

*Weather:* Weather is another important factor influencing OOHBs, a fact which has been well documented in other studies [8,40]. In Singapore, the main issues are the hot weather and frequent rain. Studies have proven that older adults are vulnerable to extreme weather [41]. In this study, a participant’s blood pressure rose to a dangerous level after walking under the hot sun. If it rains, the participants would rather stay at home unless it is necessary for them to go out. For those going out on a hot or rainy day, the provision of sheltered walkways, proximity to amenities, and connectivity (shorter distance between crossings) are very important. 

*Social contacts:* Social contact encourages older adults to go out, a fact which is supported by previous qualitative studies [8,42,43] and quantitative studies [44]. 

*High-rise, high-density:* Studies have argued that high-rise, high-density developments promote efficient and intensive land use and provide amenities and services in proximity, all of which not only lead to more walkable cities and convenient lives but also stimulate social interactions, social capital and a sense of community [45,46,47]. However, high-density developments are also associated with crowdedness, limited open spaces, congested cityscapes, a lack of privacy, an increasing alienation of inhabitants, and competition for resources and services [45,48,49]. In this study, participants expressed their preferences for larger flats and fewer neighbors sharing common corridors due to privacy, quietness, better ventilation, and coolness. As for population density, on one hand, people are attracted by facilities with more people because of vibrancy and the sense of safety [40,47]. On the other hand, crowded places inhibit older adults’ visits, especially for wheelchair users who require larger space to navigate. For older adults who enjoy drinking coffee and spending time in food places, the noisy and crowded environment may affect their chatting and discourage them from staying there for long because they may disrupt the owner’s business. In this case, the provision of free seating is important at open spaces near the eating places and markets patronized by a large number of people. 

*Affordability*: Considering that there is no universal pension system in Singapore, the affordability of shops, services, and transportation have great influence on older adults’ OOHBs, including total time spent out-of-home, number of places visited, range of activities engaged in, training, and employability. Affordability has been found to play a key role in attracting older adults to urban spaces and promoting social interactions, because commercial spaces with high-end design features tend to discourage older adults from taking a seat and having casual conversations there [50]. The cost of activities has been shown as a frequently mentioned inhibitor for participation and active ageing [51]. A study conducted in Hong Kong also indicated that affordable public transportation can compensate for unfavorable environmental factors in the neighborhood since older adults can easily travel to other neighborhoods in high-density cities [25].

*Maintenance and upgrading:* Maintenance is one of the important concerns for high-rise living since it guarantees lift operation and the cleanliness of common areas [48]. Upgrading is a unique character in Singapore’s public housing estates. It is largely subsidized by the government and is carried out every 20–30 years to enhance the living environment and services in the older estates. Generally, participants passively accept what has been offered and hold their opinions even though there are improvements they hope to see because they are aware that this top-down strategy has to be financially and strategically balanced between cost and benefits for all public housing estates in the island.

To answer the question of what factors encourage older adults to go out, this study emphasizes the necessity to consider neighborhood environmental factors, regional factors (e.g., social norms and government policy), and personal factors (e.g., health) as a whole. For example, a decision to visit a park is the combined result of one’s physical capacity, preference for an active lifestyle, the presence of parks in proximity, good pedestrian infrastructure, aesthetics, the affordability of travelling to the park and enjoying services in the park, traffic safety, good weather, the availability of companions, and health promotion policy by the government. For facilities necessary for daily living, long travel distance may prompt older adults to walk longer but may also inhibit walking if the older adults choose to use public transportation instead. 

As for the walk-along interview method, this study has shown that an unstructured walk-along interview approach can be quite beneficial because it not only discloses the implicit and unconscious aspects of the neighborhood environment but also allows the participants to share their interests, concerns, and personal histories along the way, all of which provide ample explanations for their OOHBs [12,18]. This study found that shorter period of walk-along interviews can only elicit participants’ limited feedback towards the surroundings, while those that take more than an hour can provide a more holistic picture of the older adults themselves and their ageing experiences. However, when the interview took too long, it was usually the result of getting similar stories and more waiting time. This confirms Kusenbach [18]’s conclusion that a productive time window for a go-along is about an hour to 90 min. 

Meanwhile, the walk-along approach has several limitations. First, as shared by other qualitative research methods, the extent of active participation is influenced by the participant’s age, personality and socio-economic status. This becomes more obvious in a walk-along interview because it hopes to empower the participants to gain control over the walk. In this study, those with higher socio-economic status tended to be more critical or more positive and speak continuously to take control of the whole process. Instead of wheelchair users or older adults with walking aids complaining about universal design features and long-distance to reach the amenities, those who are relatively younger or have experiences of pushing wheelchairs had stronger opinions about the built environment. This may also be due to the fact that people tend to become more tolerant and less demanding when they get older [52,53]. Second, by taking a more unstructured approach, the lengths of the routes and the range of discussion topics are hard to control. Though this study was designed to get participants’ out-of-home experiences on their daily paths between the common corridor and a neighborhood destination, the stopping and ending points depended on the participants’ willingness. Some participants shared news they read and talked about all the things they considered as issues even though they were told to talk about factors that influenced their OOHBs. This posed challenges for data analysis, such as coding and categorizing.

Except for the walk-along approach itself, this study had three limitations that should be taken into account. First, considering that Singapore is a multi-lingual, multi-racial society, the language limitations of the author may have excluded a small portion of older adults who cannot speak Mandarin or English. The language barrier may also have led to the failure of recruiting Malay participants. Second, the study was conducted in public housing estates in Yuhua East. Other neighborhoods in Singapore that follow different planning models with different sidewalk systems and service provisions may produce different results. The samples do not cover older adults living in condominiums and landed properties. Third, the sample size was very small. Though this study covered older adults with different ethnicities and different mobility capacities, the small sample size makes the results reported in this article hard to be generalized to other population. Some limitations will be addressed in the researchers’ future walk-along studies in other neighborhoods in Singapore.

## 5. Conclusions

This study concludes that access to facilities, services, and pedestrian infrastructure are important for older adults’ OOHBs. The elderly can gain a sense of familiarity via long-term residency and routine activities, which give them the confidence to go out even with reduced physical capacities. Considering Singapore’s climate, sheltered walkways, the proximity of facilities and services, and connectivity (e.g., distance between crossings) should be considered holistically. Considering the increasing number of older adults, universal design should go beyond the idea of accessibility and take a holistic view of planning accessible routes with shelters at convenient locations. Besides physical factors, social contacts and the affordability of shops and services are also important. 

Meanwhile, reflecting on the data collection process and research results, older adults’ feedback was constrained by what has been offered when living in public housing estates. Their suggestions and complaints were the results of what they saw provided by HDB in other estates. A top-down strategy and budget control make older adults passively accept the current situations. Elderly persons need to be further empowered to decide on their ideal neighborhoods for ageing-in-place. In summary, the findings from this qualitative study highlight the importance of multidimensional and contextual viewpoints to explore neighborhood environmental effects on older adults’ OOHBs. 

## Figures and Tables

**Figure 1 ijerph-16-04251-f001:**
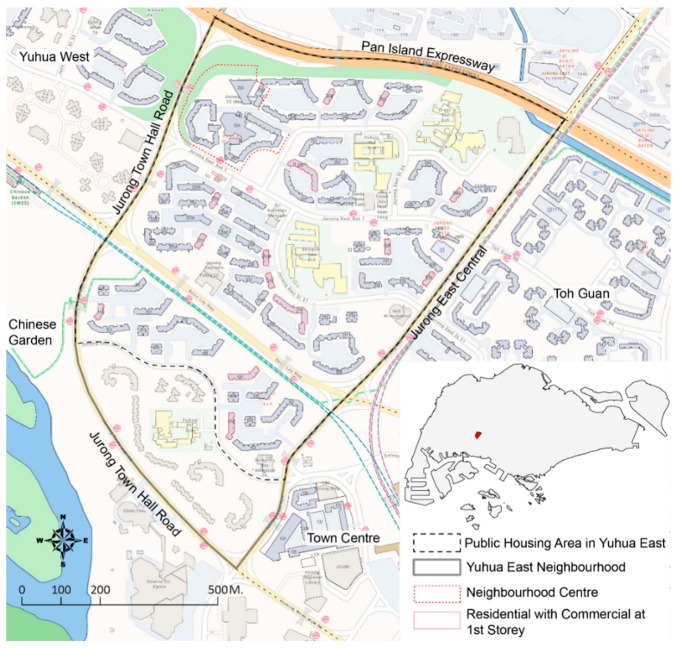
Location and site plan of the Yuhua East neighborhood.

**Figure 2 ijerph-16-04251-f002:**
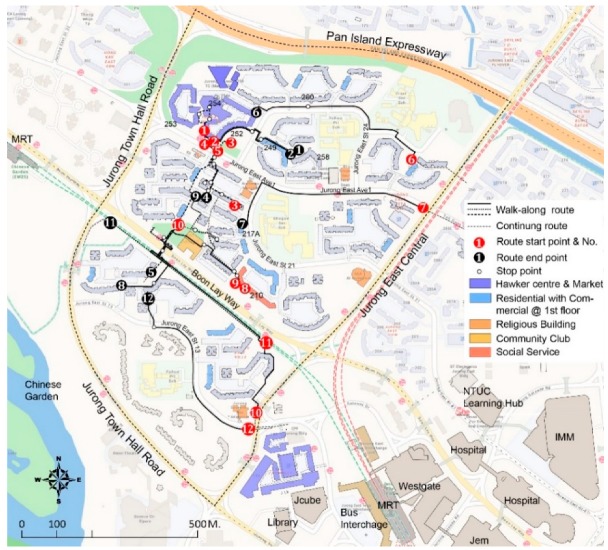
Mapping the walk-along interviews with 12 participants in Yuhua East.

**Figure 3 ijerph-16-04251-f003:**
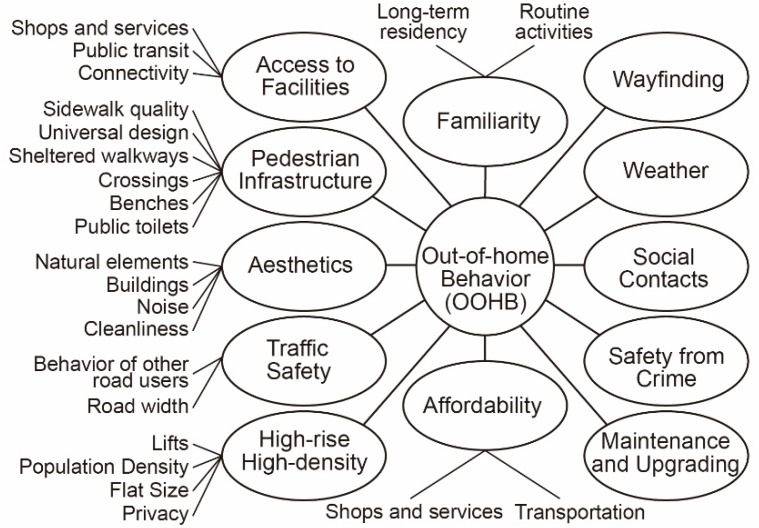
Categories and sub-categories of neighborhood environmental factors influencing older adults’ out-of-home behaviors (OOHBs).

**Figure 4 ijerph-16-04251-f004:**
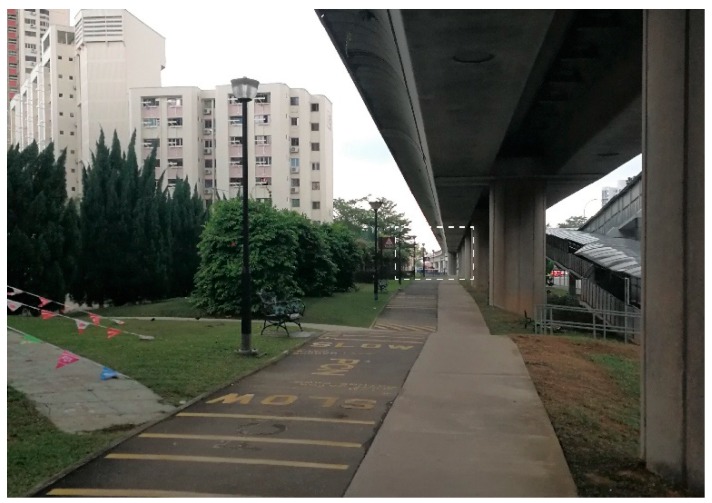
A running/cycling track and Chinese Garden MRT station in the distance.

**Figure 5 ijerph-16-04251-f005:**
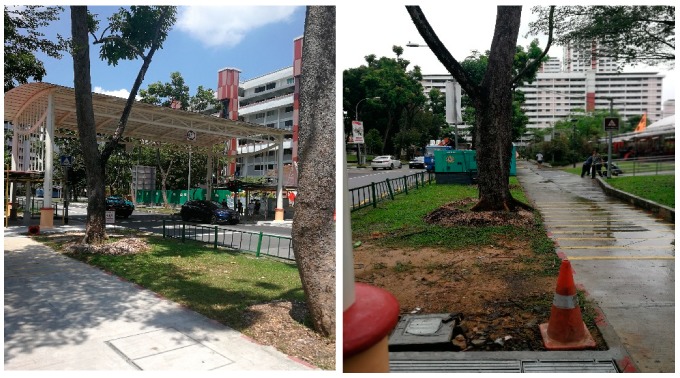
An added sheltered crossing removes the necessity of walking to the street crossing.

**Figure 6 ijerph-16-04251-f006:**
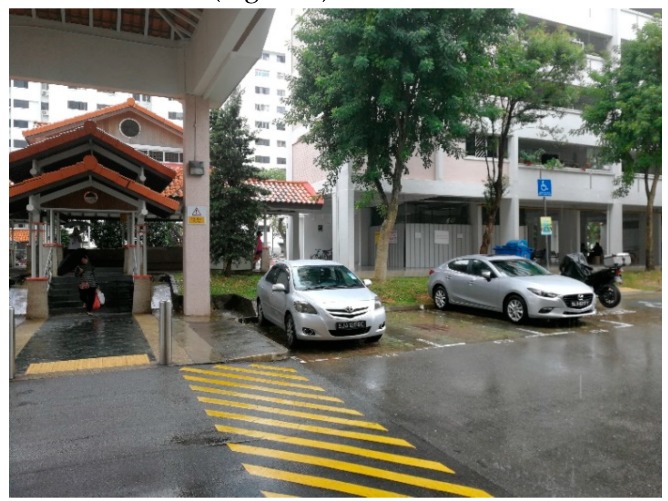
Sheltered stairs with an unsheltered slope nearby.

**Figure 7 ijerph-16-04251-f007:**
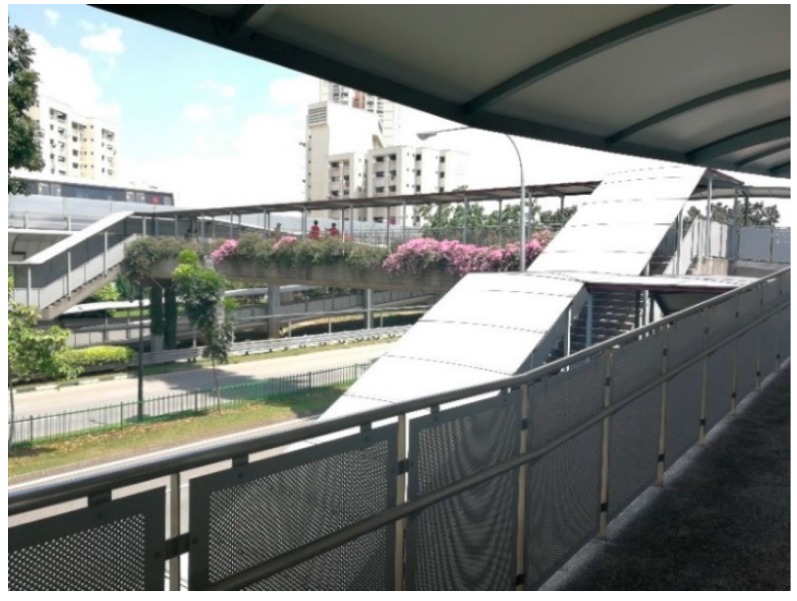
An overhead bridge with stairs and long ramps.

**Figure 8 ijerph-16-04251-f008:**
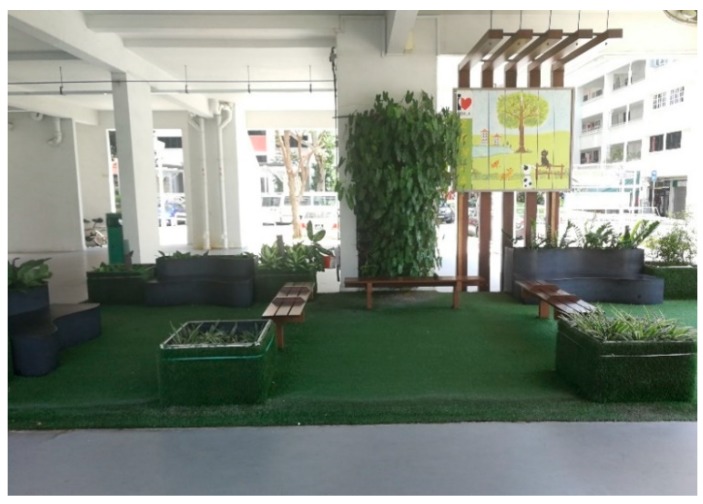
A seating area as a meeting garden in the void deck.

**Figure 9 ijerph-16-04251-f009:**
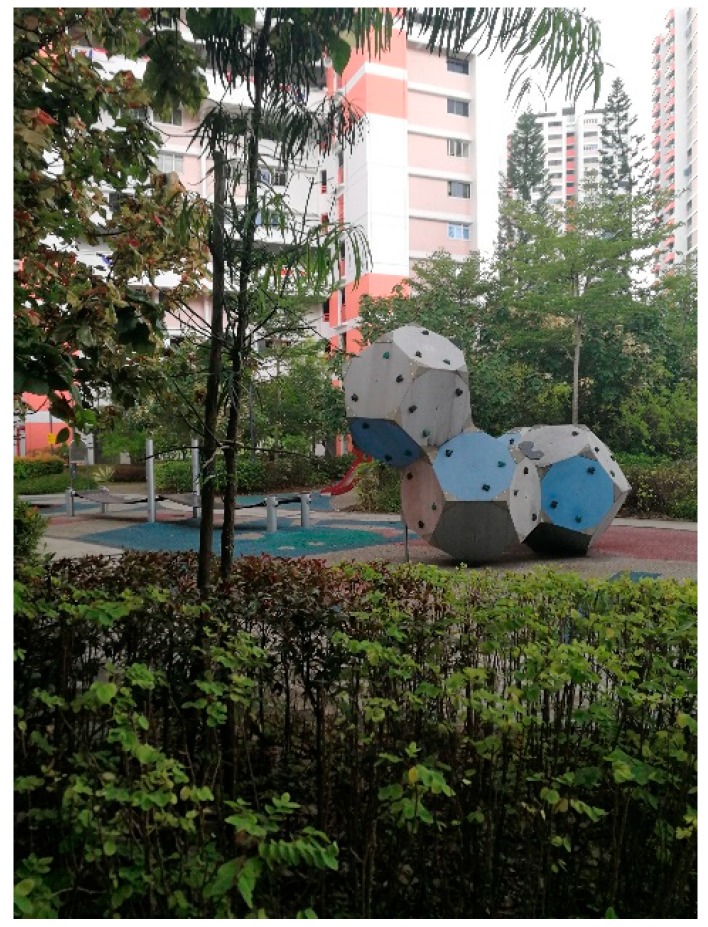
A neighborhood park with greeneries, a playground and exercise facilities.

**Figure 10 ijerph-16-04251-f010:**
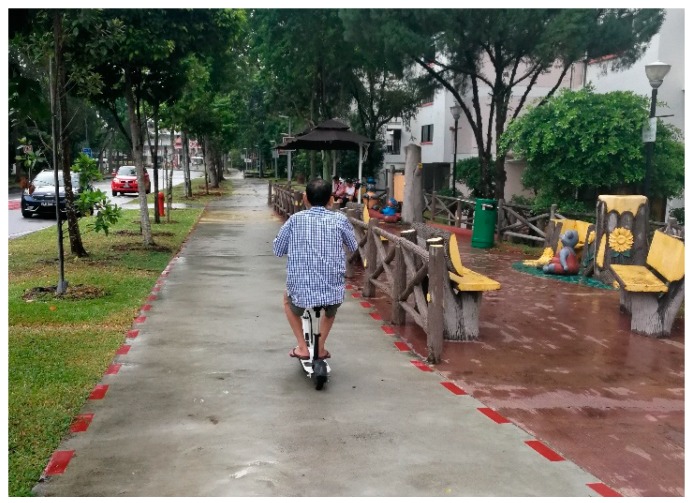
Pedestrians share the same sidewalk with those on an e-scooter.

**Figure 11 ijerph-16-04251-f011:**
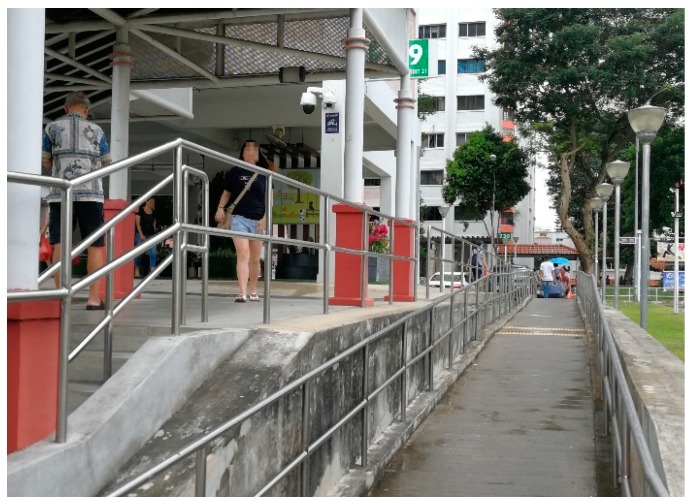
An unsheltered slope with pedestrians going back and forth to the market.

**Figure 12 ijerph-16-04251-f012:**
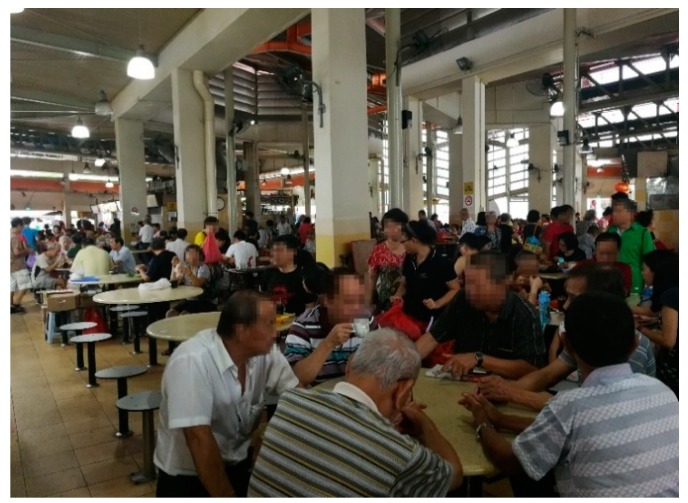
Yuhua Village Hawker Centre where people eat, drink coffee and meet friends.

**Table 1 ijerph-16-04251-t001:** Characteristics of the study participants; *N* = 12.

Characteristics		*N* (%)
Sex	Female	6 (50%)
	Male	6 (50%)
Age	55–64	2 (16.7%)
	65–74	6 (50%)
	75–84	4 (33.3%)
Ethnicity	Chinese	10 (83.3%)
	Indian	1 (8.3%)
	Other	1 (8.3%)
Mobility	Ambulant	10 (83.3%)
	Walking aids	1 (8.3%)
	Wheelchair	1(8.3%)
Years of Residency	6–10	1 (8.3%)
	11–20	2 (16.7%)
	20+	9 (75%)
Housing Type	3-Room HDB Flat	3 (25%)
	4-Room HDB Flat	3 (25%)
	5-Room HDB Flat	6 (50%)

HDB: Housing and Development Board.

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
