# Peer review of "Using Walk-Along Interviews to Identify Environmental Factors Influencing Older Adults’ Out-of-Home Behaviors in a High-Rise, High-Density Neighborhood"

_ijerph, 2019, doi:10.3390/ijerph16214251_

Round 1

Reviewer 1 Report

Overall an interesting study and it does fill a gap in the local literature.

However, the following points can be improved:

Introduction

Perhaps more emphasis can be given to the uniqueness of this study- it is carried out in Singapore, a highly urbanised Asian city-state where public housing is ubiquitous and predominantly comprised of high-rise housing. The authors should seek to draw attention that the literature mainly stems from western countries with predominantly sub-urban populations. The justification for focus on the elderly eg. ageing population should also be brought out

Methodology

I think the article suffers from insufficient detail on the methodology. Why was Yuhua East chosen out of so many public housing estates in Singapore? Again, understand that the qualitative study approach was chosen, but how was the number of 12 participants derived at? Was this sufficient for a representative study? 12 participants appears somewhat low. Additionally, I note that recruitment criteria was that participants needed to speak either English or Chinese. Singapore is a multiracial society, and most of the elderly generation may not be so fluent in English. Were minorities underrepresented in the study population (eg. Indian, Malay)? I note that only 1 of the study population was Indian and there were no Malays. The authors could have aimed for a larger sample size and a more representative distribution in terms of ethnicities. Only 2 members of the study sample have some sort of mobility impairment- again, because the perceptions of someone who is mobility-impaired would be very different from someone who is fully ambulant- if the authors were aiming for representative sampling, why was the proportion of those individuals who had mobility impairment set so low? How the start points and end points were decided is also unclear. I note that a large number of start points were clustered around Blk 252, and there is little overlap between the walking routes of different participants. My concern is that different walking routes may have different obstacles and facilities, and thus only deriving feedback from one resident who walked that particular route may be subject to bias. There was no information about how walking routes were standardised- some appear to be shorter, some appear to be longer, and the start point also does not appear to be subject to standardisation. This may cause difficulties in identifying how generalisable the study findings are. I think the methodology needs to be relooked and the authors should have increased their study sample to include larger proportions of minority ethnicities and mobility-impaired individuals

Others

The discussion should be more structured and organised according to general themes.

Referencing needs to be improved. Throughout the paper, "Error! Reference source not found" pops up quite a lot. The overall English also can be improved upon.

Author Response

Dear Reviewer,

We would like to thank the reviewer for deep and thorough review and for the insightful comments and constructive suggestions. The followings are our point-by-point responses:

Point 1: Overall an interesting study and it does fill a gap in the local literature.

Response 1: We are grateful to the reviewer for the encouraging comments.

Point 2: Introduction. Perhaps more emphasis can be given to the uniqueness of this study- it is carried out in Singapore, a highly urbanised Asian city-state where public housing is ubiquitous and predominantly comprised of high-rise housing. The authors should seek to draw attention that the literature mainly stems from western countries with predominantly sub-urban populations. The justification for focus on the elderly eg. ageing population should also be brought out. 

Response 2: Thank you for pointing out the area for improvement. The uniqueness of the study site and ageing population in Singapore have been added as “Singapore is a highly urbanised Asian city-state where more than 80% of the residents live in high-rise high-density public housing estates developed by Housing and Development Board (HDB). It is ageing rapidly where the proportion of residents aged 65 years and over is 13.7% and those aged 55 and over is 28.1%” (see lines 63-67). We have also added “western countries with predominantly sub-urban populations” to the introduction section in lines 59-60.

Point 3: Methodology. I think the article suffers from insufficient detail on the methodology. Why was Yuhua East chosen out of so many public housing estates in Singapore?

Response 3: We are grateful for the detailed comments about methodology.  We hope to address the comments in Response 3-9. For why Yuhua East was chosen, first, we have added “It is a typical example of HDB new town developments around 1980s, following “New Town Structural Model” with amenities planned at three hierarchies from a town centre, to neighbourhood centres and precinct centres” in lines 72-74. Although younger generations of HDB new town developments focus more on identity and character, they are planned based on the “New Town Structural Model”.  We did not mention another reason in the manuscript considering its relevance and the length of the manuscript. As this manuscript is part of an on-going study with a larger research scope, other three neighbourhoods are selected for future studies, developed at different time periods with different ageing population percentages.

Point 4:  Methodology.  Again, understand that the qualitative study approach was chosen, but how was the number of 12 participants derived at? Was this sufficient for a representative study? 12 participants appears somewhat low.

Response 4: Thank you for the comments. We agree with the reviewer that the sample size is small and have acknowledged this as a limitation in the discussion section in lines 652-654. To answer how was the number of 12 participants derived at, first, we have made the criteria of “purposeful convenience sampling” clearer. We have deleted “To be more inclusive, the sample includes people belong to different age groups, with different gender, ethnicities, mobility, education” and have added “Although older adults are generally defined as those above 65 years and over, this study included those aged 55-64 considering the fact that Singapore is ageing rapidly and their opinions are valuable for future developments. The percentage of participants aged 55-64 was expected to be less than 20%. Selected samples were also expected to cover similar numbers of male and female, including Chinese, Malay, Indian and Others (see lines 115-120).” Second, we conducted a literature review about sample sizes for qualitative studies. Sandelowski suggests that qualitative sample sizes of 10 may be adequate for sampling among a homogenous population [1]. As “data saturation” is frequently mentioned as a justification for sample sizes in qualitative studies, another article investigates actual theoretical saturation and finds data saturation starting to become evident at 6 in-depth interviews and definitely evident at 12 in-depth interviews among a relatively homogeneous population [2]. Admittedly, older adults in Singapore are not a homogeneous group, however the participants all live in public housing estates and the intention of the research is to gain opinions about how neighbourhood environment influences out-of-home behaviors (OOHBs). During data collection, expect from one participant from Myanmar mentioned she went to an Indian shop to find favourite food, we did not find clear differences between Chinese and people with other ethnicities in terms of feedbacks towards neighourhood environment and codes. Not much happened to the number of codes. We also find other qualitative studies use 10 or 12 as the sample size.  For example, one study using go-along interviews to uncover landscape values and micro-geographies of meanings had 10 samples [3]. Another study recruited 12 participants to understand older adults’ adaptive environmental behaviours [4]. Third, the number of 12 participants was derived based on compromising considerations because we will conduct walk-along interviews in other three neighbourhoods in the future. Sandelowski suggests 50 interviews is a large sample for a qualitative study [1].  On one hand, we hope there are relatively enough samples in one neighbourhood to be as representative as possible. On the other hand, we hope to control the total sample size below 50 considering our limited resources. We have also added “Some limitations will be addressed in the researchers’ future walk-along studies in the other neighbourhoods in Singapore” in lines 654-656.

Point 5:  Methodology. Additionally, I note that recruitment criteria was that participants needed to speak either English or Chinese. Singapore is a multiracial society, and most of the elderly generation may not be so fluent in English.

Response 5: The recruitment criteria is due to the language barrier of the researcher. We have explained the researcher’s language background in lines 114-115. We have also added this limitation to the discussion section in lines 646-649.

Point 6:  Were minorities underrepresented in the study population (eg. Indian, Malay)? I note that only 1 of the study population was Indian and there were no Malays. The authors could have aimed for a larger sample size and a more representative distribution in terms of ethnicities.

Response 6:  Thank you for highlighting the issue of the underrepresented minorities. Although the total percentage of the minorities of the study population relatively matches the demographics of Yuhua East (information have been added in lines 81-85), we admit that we failed to recruit Malays and have added “The researcher failed to recruit Malay participants although attempts were made” to the result section in lines 139-140 and have added this limitation to the discussion section in line 648. This is probably due to the researcher’s language barriers. During data collection, we did make several attempts and even went to the mosque within Yuhua East because we did not succeed at other amenities and open spaces. Although 6 attempts were made at the entrance of the mosque, they were either younger or do not live within the neighbourhood. In our future studies, we will include more minorities to address this limitation.

Point 7:  Methodology. Only 2 members of the study sample have some sort of mobility impairment- again, because the perceptions of someone who is mobility-impaired would be very different from someone who is fully ambulant- if the authors were aiming for representative sampling, why was the proportion of those individuals who had mobility impairment set so low?

Response 7: Thank you for the comment. For those with mobility impairments, according to the latest National Survey of Senior Citizens in Singapore [5],  among those aged 75 years and over,  86.5% are ambulant. The number is larger for younger age-groups. As explained in Response 4 about the criteria of “purposeful convenience sampling”, we did not approach older adults with mobility limitations deliberately and we were able to recruit 2 participants as a result. We agree with the reviewer that the perceptions of someone who is mobility-impaired would be very different from someone who is fully ambulant. Future studies focusing more on those with mobility impairments will be meaningful, however, it is beyond the scope of this manuscript.

Point 8:  Methodology. How the start points and end points were decided is also unclear. I note that a large number of start points were clustered around Blk 252, and there is little overlap between the walking routes of different participants. My concern is that different walking routes may have different obstacles and facilities, and thus only deriving feedback from one resident who walked that particular route may be subject to bias. There was no information about how walking routes were standardised- some appear to be shorter, some appear to be longer, and the start point also does not appear to be subject to standardisation. This may cause difficulties in identifying how generalisable the study findings are.

Response 8: We really appreciate this comment. Some specific details of the walk-along method have been added as “The participants were recruited at amenities and open spaces within the boundary of Yuhua East. After explaining the research purposes and procedures to the participants, the researcher walked together with the participant and asked questions along the way from a neighbourhood destination to the common corridor of the participant’s home, or the other way around. The walk would stop or begin, normally at the void decks (vacant spaces on the ground floor of HDB flats), if the participant felt uncomfortable. In this case, information about the missing parts of the route would be collected without walking-along (lines 102-108).” We hope this information can explain the standard we apply in terms of what kinds of information this manuscript hopes to collect. We have also added a limitation in lines 638 -642 as “by taking a more unstructured approach, the lengths of the routes and the range of discussion topics are hard to control. Although this study is designed to get participants’ out-of-home experiences on their daily paths between the common corridor and a neighbourhood destination, the stopping and ending points depend on the participants’ willingness.”

Blk 252 is located at the neighbourhood centre where there are larger number of older adults, which is why it is the starting point of many routes. As for biases, we agree with the reviewer that one participant’s opinion may have biases. During data collection, we do not feel biases are strong, probably due to the homogeneity of HDB public housing estates. The provision of facilities follows a certain standard and “The Code on Accessibility” has been applied broadly in HDB housing estates. Also, the categories of neighbourhood environmental factors are resulted from 12 participants as a whole. Although we agree with the reviewer that more standardised routes will make the manuscript more generalisable, the walk-along routes were not controlled because we hope to collect information of older adults’ daily routes. We also hope to walk on different routes to cover different areas so that we can get a more holistic understanding of Yuhua East.

Point 9: Methodology.  I think the methodology needs to be relooked and the authors should have increased their study sample to include larger proportions of minority ethnicities and mobility-impaired individuals.

Response 9: Thank you again for the detailed comments about methodology. We hope we have addressed the reviewer’s concerns in Response 3-8. We agree with the reviewer that the small sample size is a limitation and we will address it in future studies.  

Point 10: The discussion should be more structured and organised according to general themes.

Response 10: Thank you for the suggestion. The discussion section has been updated to be more structured and organised. 

Point 11: Referencing needs to be improved. Throughout the paper, "Error! Reference source not found" pops up quite a lot.

Response 11: Referencing has been improved as suggested.

Point 12: The overall English also can be improved upon.

Response 12: As suggested, a professional editing and proofreading service has been used before submitting the revised version.

References

Sandelowski, M., Sample Size in Qualitative Research. Research in Nursing & Health, 1995. 18(2): p. 179-183. Guest, G., A. Bunce, and L. Johnson, How Many Interviews Are Enough?: An Experiment with Data Saturation and Variability. Field Methods, 2006. 18(1): p. 59-82. Bergeron, J., S. Paquette, and P. Poullaouec-Gonidec, Uncovering Landscape Values and Micro-geographies of Meanings with the Go-along Method. Landscape and Urban Planning, 2014. 122: p. 108-121. Lien, L.L., C.D. Steggell, and S. Iwarsson, Adaptive Strategies and Person-environment Fit among Functionally Limited Older Adults Aging in Place: A Mixed Methods Approach. International Journal of Environmental Research and Public Health, 2015. 12(9): p. 11954-11974. Kang, S.H., E.S. Tan, and M.T. Yap, National Survey of Senior Citizens 2011. 2013, Singapore: Institute of Policy Studies, National University of Singapore.

Reviewer 2 Report

Thank you for the opportunity to review this manuscript. Please find below feedback intended to help strengthen the manuscript.

Avoid the term “the elderly” in your manuscript. As of January 2018, major medical and healthcare journals have adopted the American Medical Association Manual of Style, which recommends avoiding the terms “elders” and “seniors,” and instead using “older adult” in reference to people above the age of 65. CDC defines an older adult as someone age 60 or older, but there is no general international agreement on the age at which a person becomes old. Your study may have used a different definition because this definition does not adapt well to the situation in Singapore. In that case, you must state the reason for using a different definition than the commonly used one (60+ or 65+ years). It might be due to some cultural differences, different retirement age, or any other reason. In line 98, authors say “To be more inclusive, the sample includes people belong to different age groups, with different gender, ethnicities, mobility, education.” How do you justify such a small sample size then? When there are too many uncontrolled variables in a study, the sample size needs to be much larger. There are issues with developed categories and sub-categories that need to be fixed. For example, noise is not an aesthetic quality; cleanliness relates to maintenance, not aesthetics; the public toilet is part of the urban construction, not a walking facility. The manuscript needs major editing throughout. Authors have missed words like “the”, used verbs with present tense to describe past events, missed “s” for plural words, and used incorrect prepositions. Authors need to use a professional editing and proofreading service before submitting the revised version. Replace “&” with “and” throughout the manuscript Introduce “hawker center” in a footnote. Correct the reference errors (e.g. in lines 174, 121, 122) In line 68, change “research” to “researchers.

Author Response

Dear reviewer, 

We would like to thank the reviewer for deep and thorough review and for the insightful comments and constructive suggestions. The followings are our point-by-point responses:

Point 1: Avoid the term “the elderly” in your manuscript. As of January 2018, major medical and healthcare journals have adopted the American Medical Association Manual of Style, which recommends avoiding the terms “elders” and “seniors,” and instead using “older adult” in reference to people above the age of 65. CDC defines an older adult as someone age 60 or older, but there is no general international agreement on the age at which a person becomes old. Your study may have used a different definition because this definition does not adapt well to the situation in Singapore. In that case, you must state the reason for using a different definition than the commonly used one (60+ or 65+ years). It might be due to some cultural differences, different retirement age, or any other reason.

Response 1: Thank you for your comments and expertise. As suggested, “the elderly” has been changed to “older adults” in our manuscript.   We have also added the rationale for our definition of “older adults” in lines 115-117 as “Although older adults are generally defined as those above 65 years and over, this study included those aged 55-64 considering the fact that Singapore is ageing rapidly and their opinions are valuable for future developments.”

Point 2: In line 98, authors say “To be more inclusive, the sample includes people belong to different age groups, with different gender, ethnicities, mobility, education.” How do you justify such a small sample size then? When there are too many uncontrolled variables in a study, the sample size needs to be much larger.

Response 2: Thank you for highlighting the issue of sample size. We agree with the reviewer that the sample size is small and have acknowledged this as a limitation in the discussion section in lines 652-654. We have deleted “To be more inclusive, the sample includes people belong to different age groups, with different gender, ethnicities, mobility, education”. To be clearer about the criteria of “purposeful convenience sampling”, we have added “The percentage of the participants aged 55-64 was expected to be less than 20%. Selected samples were also expected to cover similar number of male and female, including Chinese, Malay, Indian and Others” (see lines 118-120).  As for why the sample size is 12, first, we conducted a literature review about sample sizes for qualitative studies. One study suggested that qualitative sample sizes of 10 may be adequate for sampling among a homogenous population [1]. As “data saturation” is frequently mentioned as a justification for sample sizes in qualitative studies, another article investigates actual theoretical saturation and finds data saturation starting to become evident at 6 in-depth interviews and definitely evident at 12 in-depth interviews among a relatively homogeneous population [2]. Admittedly, older adults in Singapore are not a homogeneous group, however the participants all live in public housing estates and the intention of the research is to gain opinions about how neighbourhood environment influences out-of-home behaviours (OOHBs). During data collection, expect from one participant from Myanmar mentioned she went to an Indian shop to find favourite food, we did not find clear differences between Chinese and people with other ethnicities in terms of feedbacks towards neighourhood environment and codes. Not much happened to the number of codes. Second, the number of 12 participants was derived based on compromising considerations because we will conduct walk-along interviews in other three neighbourhoods in the future. Sandelowski suggests 50 interviews is a large sample for a qualitative study [1].  On one hand, we hope there are relatively enough samples in one neighbourhood to be as representative as possible. On the other hand, we hope to control the total sample size below 50 considering our limited resources. We have also added “Some limitations will be addressed in the researchers’ future walk-along studies in the other neighbourhoods in Singapore” in lines 654-656.

Point 3: There are issues with developed categories and sub-categories that need to be fixed. For example, noise is not an aesthetic quality; cleanliness relates to maintenance, not aesthetics; the public toilet is part of the urban construction, not a walking facility.

Response 3: Thank you for your comments. We have reconsidered the developed categories and sub-categories. As mentioned in lines 134-135, categories and subcategories were derived based on previous studies for ease of comparisons. For noise and cleanliness, we still include them under “aesthetics” for two reasons. First, they are included in “aesthetics” in the article by Van Cauwenberg et al. (reference no. 8). Second, Thomas Leddy states some grander aesthetic qualities found in everyday life such as “neat” and “clean" are left out in the literature of aesthetics [3]. During data collection, we found that when the participants describe their neighbourhoods as “nice” and/or “beautiful”, they tended to talk about cleanness and quietness. For public toilet, it is listed under “Pedestrian/cycling Infrastructure” in the review article by Cerin et al. (reference no. 6). Considering older adults tend to use toilet more frequently and many older adults walk longer distances (e.g., 10000 steps) for exercises, the presence of public toilet does influence their OOHBs such as walking.  We have changed “walking facilities” with “pedestrian infrastructure” in the revised manuscript.

Point 4: The manuscript needs major editing throughout. Authors have missed words like “the”, used verbs with present tense to describe past events, missed “s” for plural words, and used incorrect prepositions. Authors need to use a professional editing and proofreading service before submitting the revised version.

Response 4: As suggested, a professional editing and proofreading service has been used before submitting the revised version.

Point 5: Replace “&” with “and” throughout the manuscript.

Response 5: As suggested, we have replaced “&” with “and” throughout the manuscript.

Point 6: Introduce “hawker center” in a footnote.

Response 6: Thank you for the comment. Hawker centre has been explained in line 164. Considering the explanation is quite short, it is included in the main text.

Point 7: Correct the reference errors (e.g. in lines 174, 121, 122)

Response 7: As suggested, references errors have been corrected. 

Point 8: In line 68, change “research” to “researchers. 

Response 8: Thank for pointing out this. We have changed it as suggested.   

References

Sandelowski, M., Sample Size in Qualitative Research. Research in Nursing & Health, 1995. 18(2): p. 179-183. Guest, G., A. Bunce, and L. Johnson, How Many Interviews Are Enough?: An Experiment with Data Saturation and Variability. Field Methods, 2006. 18(1): p. 59-82. Leddy, T., Everyday Surface Aesthetic Qualities: "Neat," "Messy," "Clean," "Dirty". The Journal of Aesthetics and Art Criticism, 1995. 53(3): p. 259-268.

Reviewer 3 Report

The article presents a qualitative study of environmental factors associated with enabling or preventing older adults in an urban setting from interacting with their neighborhood environments.  I commend the authors on a very interesting study presented well.  I believe this article to be a valuable addition to the current literature, especially for the methods that it applies to this specific research question.  However, a few changes are needed and points clarified before it is suitable for dissemination. 

Overall comment:

Throughout the manuscript, there is mention of a deductive approach to this study, with mention that a categorization matrix was developed based on an earlier study. However, with discussions of using unconstrained matrices, creation of new categories, and unstructured interviewing approaches, the approach used in this study appears more inductive.  If this is not the case, a discussion in the methods section towards the end of data collection and analysis about the categories used to structure the process, along with clarification about the structure of the matrix, would help improve the clarity of the analytical method.

Introduction:

Line 39 states that there are few quantitative studies. There are a number of studies in the literature, which I agree with the authors largely focus on physical activity.  However, “few” is not a fair characterization of the literature.  Please revise this statement On a similar note, line 53 again states “few,” which I do not believe is a fair adjective to describe the current volume of applicable literature.

Materials and Methods:

The discussion of the walk-along method is very useful, however, I found myself wondering what the method entails exactly. What actually happens?  I imagine it is very similar to the title.  A researcher walks along with a subject and conducts a structured, semi-structured, or unstructured interview.  However, more specific details about this method and how it is performed would further strengthen this subsection. On line 77-81, the authors discuss the researcher’s involvement in the method and the degree of structure of the interview. I was expecting to see what approach was used in this study.  It is stated later that a more unstructured approach was used, which is valid.  However, this information would also be beneficial to the reader here in section 2.1. In section 2.2, additional information about the context of the neighborhood is needed. Specifically missing are additional socio-economic and ethnic demographics of the residents of this area, if available. Section 2.3, lines 97-98, the age of subjects is stated as 55 years old. I believe the authors intend to say 55 years old or older, but I’m not certain. Could this be clarified? In lines 108-110, there is discussion of what types of information are gathered during the interviews. Later, a map of routes is also presented.  How was this information collected to create this map?  GPS tracking?  Collected as part of the field notes? Line 114 – the translation was performed by the researchers. This would assume that the researchers (at least in part) were bilingual.  If so, this should be made explicit earlier. 

Results

Line 114 mentions a hawker center and market – a small phrase describing what this is would benefit international audiences, as I was unfamiliar with this specific term. Section 3.11 – the term legibility may be better suited in this field by renaming to “wayfinding.”

Discussion

The paragraph starting on line 606 begins with a limitation attributed to the walk-along method itself. While the latter limitations mentioned in this paragraph are related to the method, the extent of participation is stated as being due to age, personality, and education.  This may be the case for other qualitative methods as well, and might be better located in the next paragraph.

Thank you for the opportunity to review this manuscript, and I look forward to receiving and discussing comments from the authors.

Author Response

Dear Reviewer, 

We would like to thank the reviewer for deep and thorough review and for the insightful comments and constructive suggestions. The followings are our point-by-point responses:

Point 1: The article presents a qualitative study of environmental factors associated with enabling or preventing older adults in an urban setting from interacting with their neighborhood environments.  I commend the authors on a very interesting study presented well.  I believe this article to be a valuable addition to the current literature, especially for the methods that it applies to this specific research question. 

Response 1: We are grateful to the reviewer for the encouraging comments.

Point 2: Overall comments. Throughout the manuscript, there is mention of a deductive approach to this study, with mention that a categorization matrix was developed based on an earlier study. However, with discussions of using unconstrained matrices, creation of new categories, and unstructured interviewing approaches, the approach used in this study appears more inductive.  If this is not the case, a discussion in the methods section towards the end of data collection and analysis about the categories used to structure the process, along with clarification about the structure of the matrix, would help improve the clarity of the analytical method.

Response 2: Thank you for your suggestion and expertise. We agree with the reviewer that this study appears to be more inductive.  We have changed it accordingly in lines 134-135.   

Point 3: Introduction. Line 39 states that there are few quantitative studies. There are a number of studies in the literature, which I agree with the authors largely focus on physical activity.  However, “few” is not a fair characterization of the literature.  Please revise this statement On a similar note, line 53 again states “few,” which I do not believe is a fair adjective to describe the current volume of applicable literature.

Response 3: Thank you for pointing this out. “Few” in line 39 has been revised as “a number of”.   “Few” in line 54 has been replaced with “a small number of”.

Point 4: Materials and Methods. The discussion of the walk-along method is very useful, however, I found myself wondering what the method entails exactly. What actually happens?  I imagine it is very similar to the title.  A researcher walks along with a subject and conducts a structured, semi-structured, or unstructured interview.  However, more specific details about this method and how it is performed would further strengthen this subsection. In lines 77-81, the authors discuss the researcher’s involvement in the method and the degree of structure of the interview. I was expecting to see what approach was used in this study.  It is stated later that a more unstructured approach was used, which is valid.  However, this information would also be beneficial to the reader here in Section 2.1.

Response 4: Thank you for the important comments. First, we have changed the orders of Section 2.1 and Section 2.2 to introduce the study area first and then talk about the walk-along method. Second, we have adjusted the structure of the manuscript and discuss the researcher’s involvement immediately after introducing the walk-along method (see line 101). Third, more details have been added to the section in lines 102-108 as “The participants were recruited at amenities and open spaces within the boundary of Yuhua East. After explaining the research purposes and procedures to the participants, the researcher walked together with the participant and asked questions along the way from a neighbourhood destination to the common corridor of the participant’s home, or the other way around. The walk would stop or begin, normally at the void decks (vacant spaces on the ground floor of HDB flats), if the participant felt uncomfortable. In this case, information about the missing parts of the route would be collected without walking-along.”

Point 5: Materials and Methods. In section 2.2, additional information about the context of the neighborhood is needed. Specifically missing are additional socio-economic and ethnic demographics of the residents of this area, if available.

Response 5: We appreciate your comments.  In lines 81-86, we have added additional information including socio-economic and ethnic demographics of the Yuhua East residents.

Point 6: Materials and Methods.  Section 2.3, lines 97-98, the age of subjects is stated as 55 years old. I believe the authors intend to say 55 years old or older, but I’m not certain. Could this be clarified?

Response 6: We agree with the reviewer that we intended to say “55 years old and over” and have changed it accordingly.

Point 7: Materials and Methods. In lines 108-110, there is discussion of what types of information are gathered during the interviews. Later, a map of routes is also presented.  How was this information collected to create this map?  GPS tracking?  Collected as part of the field notes?

Response 7: Thank you for the comment. In lines 125-126, we have added “The routes were hand drawn on printed maps, compiled and digitalised later”.

Point 8: Materials and Methods. Line 114 – the translation was performed by the researchers. This would assume that the researchers (at least in part) were bilingual.  If so, this should be made explicit earlier. 

Response 8: Thank you for pointing this out. In lines 114-115, we have added “The researcher’s mother language is Mandarin Chinese, with Test of English as a Foreign Language® (TOEFL®) test score of 106” to explain the researcher’s language background.

Point 9:   Results. Line 114 mentions a hawker center and market – a small phrase describing what this is would benefit international audiences, as I was unfamiliar with this specific term. Section 3.11 – the term legibility may be better suited in this field by renaming to “wayfinding.”.

Response 9: Thank you for the comments. In lines 164-165, a small phrase describing hawker center has been added. As suggested, the term “legibility” has been renamed to “wayfinding”.

Point 10:  Discussion. The paragraph starting on line 606 begins with a limitation attributed to the walk-along method itself. While the latter limitations mentioned in this paragraph are related to the method, the extent of participation is stated as being due to age, personality, and education.  This may be the case for other qualitative methods as well, and might be better located in the next paragraph.

Response 10: Thank you for the comment. We agree with the reviewer that the limitation of “the extent of participation is influenced by age, personality, and education (changed as secio-economic status)” is shared by studies using other qualitative methods. However, for walk-along interviews, we feel this tendency is stronger because it hopes to empower the participants to take control over the walk. We have rewritten the sentences as “First, as shared by other qualitative research methods, the extent of active participation is influenced by the participant’s age, personality and socio-economic status. This becomes more obvious in a walk-along interview …” (see lines 629-631)

We hope you find our responses and the revised manuscript satisfactory. We look forward to hearing from you in due time regarding our submission and to respond to any further questions and comments you may have.

Reviewer 4 Report

This paper is a valuable contribution to measure environmental factors for the elderly's walking for transportation.

There are some comments that allow improving the work:

1.P.3, line 97: The sample is small. How can the author prove these participants were representative in this study area? What criteria are the study’s “purposeful convenience sampling” ?

2.P.4, line 124: The table of descriptive statistics is absent. The author should clarify the sample is representative by the table of descriptive statistics.

3.P.5, line 127: The table of “percentages of participants that discussed on neighborhood environmental factors” should be listed. The author should show the number and ratio to prove the categories and sub-categories are meaningful.

4.P.13, line372~396: Factor of “High-rise High Density” is not appeared in Figure 1 but discussed in results.

5.P.14~15, line438~494: Factor of “Social Norms” is not appeared in Figure 1 but discussed in results.

Author Response

Dear Reviewer,

We would like to thank the reviewer for deep and thorough review and for the insightful comments and constructive suggestions. The followings are our point-by-point responses:

Point 1: P.3, line 97: The sample is small. How can the author prove these participants were representative in this study area? What criteria are the study’s “purposeful convenience sampling”?

Response 1: Thank you for the comments. We have added the criteria of “purposeful convenience sampling” in lines 118-120 as “The percentage of participants aged 55-64 was expected to be less than 20%. Selected samples were also expected to cover similar numbers of male and female, including Chinese, Malay, Indian and Others.” We have also explained in lines 115-117 why we include those aged 55-64 as “Although older adults are generally defined as those above 65 years and over, this study included those aged 55-64 considering the fact that Singapore is ageing rapidly and their opinions are valuable for future developments.”

As a qualitative study, our purpose is to understand older adults’ perceptions about neighbourhood environment. As shared by other qualitative studies, we admit that the small sample size makes the results reported in the manuscript hard to be generalized to other population and have acknowledged this as a limitation in lines 652-654. As for why the sample size is 12, first, we conducted a literature review about sample sizes for qualitative studies. One study suggested that qualitative sample sizes of 10 may be adequate for sampling among a homogenous population [1]. As “data saturation” is frequently mentioned as a justification for sample sizes in qualitative studies, another article investigates actual theoretical saturation and finds data saturation starting to become evident at 6 in-depth interviews and definitely evident at 12 in-depth interviews among a relatively homogeneous population [2]. Admittedly, older adults in Singapore are not a homogeneous group, however the participants all live in public housing estates and the intention of the research is to gain opinions about how neighbourhood environment influences out-of-home behaviours (OOHBs). During data collection, expect from one participant from Myanmar mentioned she went to an Indian shop to find favourite food, we did not find clear differences between Chinese and people with other ethnicities in terms of feedbacks towards neighourhood environment and codes. Not much happened to the number of codes. We also find other qualitative studies use 10 or 12 as the sample size [3, 4].  Second, the number of 12 participants was derived based on compromising considerations because we will conduct walk-along interviews in three other neighbourhoods in the future. Sandelowski suggests 50 interviews is a large sample for a qualitative study [1].  On one hand, we hope there are relatively enough samples in one neighbourhood to be as representative as possible. On the other hand, we hope to control the total sample size below 50 considering our limited resources. We have also added “Some limitations will be addressed in the researchers’ future walk-along studies in the other neighbourhoods in Singapore” in lines 654-656.

Point 2: P.4, line 124: The table of descriptive statistics is absent. The author should clarify the sample is representative by the table of descriptive statistics.

Response 2: Thank you for pointing this out. We have replaced Table 1 with the table of descriptive statistics. See P. 4, line 143.

Point 3: P.5, line 127: The table of “percentages of participants that discussed on neighborhood environmental factors” should be listed. The author should show the number and ratio to prove the categories and sub-categories are meaningful.

Response 3: Thank you for your comment and expertise. We have added the table of “Numbers and percentages of participants that discussed an environmental category”. See lines 154-155 and Table A1 in the Appendix.

Point 4: P.13, line372~396: Factor of “High-rise High Density” is not appeared in Figure 1 but discussed in results.

Response 4: Thank you for the comment. We found the referencing was wrong. In the revised manuscript, we believe the figure that the reviewer refers to is Figure 3. “High-rise High-density” has been included in the figure.

Point 5: P.14~15, line438~494: Factor of “Social Norms” is not appeared in Figure 1 but discussed in results.

Response 5: Thank you for pointing this out. We have mentioned in lines 155-157 “By taking an unstructured approach, some important categories were uncovered that went beyond environmental factors, including social norms (social attitudes towards ageing, gender difference, multi-ethnic country), government policy and health.” As Figure 3 (previously Figure 1) is about categories and sub-categories of neighbourhood environmental factors, we did not include social norms in the figure.

References

Sandelowski, M., Sample Size in Qualitative Research. Research in Nursing & Health, 1995. 18(2): p. 179-183. Guest, G., A. Bunce, and L. Johnson, How Many Interviews Are Enough?: An Experiment with Data Saturation and Variability. Field Methods, 2006. 18(1): p. 59-82. Bergeron, J., S. Paquette, and P. Poullaouec-Gonidec, Uncovering Landscape Values and Micro-geographies of Meanings with the Go-along Method. Landscape and Urban Planning, 2014. 122: p. 108-121. Lien, L.L., C.D. Steggell, and S. Iwarsson, Adaptive Strategies and Person-environment Fit among Functionally Limited Older Adults Aging in Place: A Mixed Methods Approach. International Journal of Environmental Research and Public Health, 2015. 12(9): p. 11954-11974.

Round 2

Reviewer 1 Report

The authors have responded to my queries appropriately.

No further input.

Reviewer 2 Report

Thanks for addressing the comments.